# Quantization-Free Autoregressive Action Transformer

**Ziyad Sheebaelhamd**
University of Tübingen

**Michael Tschannen**
Google DeepMind

**Michael Muehlebach**
Max Planck Institute for Intelligent Systems

**Claire Vernade**
University of Tübingen

## Abstract

Current transformer-based imitation learning approaches introduce discrete action representations and train an autoregressive transformer decoder on the resulting latent code. However, the initial quantization breaks the continuous structure of the action space thereby limiting the capabilities of the generative model. We propose a quantization-free method instead that leverages Generative Infinite-Vocabulary Transformers (GIVT) as a direct, continuous policy parametrization for autoregressive transformers. This simplifies the imitation learning pipeline while achieving state-of-the-art performance on a variety of popular simulated robotics tasks. We enhance our policy roll-outs by carefully studying sampling algorithms, further improving the results.

## 1 Introduction

Generative modeling lies at the heart of modern machine learning, enabling the production of outputs that adhere to user-specified objectives and constraints by navigating high-dimensional spaces of potential outcomes. Fundamentally, this process can be understood as a form of search where naive or exhaustive strategies are computationally intractable due to the size of the space. We consider imitation learning for robotic systems as an extreme example, where the search space has uncountable cardinality and a complex structure arising from nonlinear dynamics, interactions with the environment and the fundamental multimodality in human or robot behavior. In many real-world robotics applications, the control signals involved are continuous in nature [33, 59, 28]. Current state-of-the-art methods for modeling policies in continuous control tasks broadly fall into two categories: autoregressive transformer models [26, 45] and diffusion models [7, 54, 40], which will be discussed in the next two paragraphs.

Existing autoregressive policies, on the one hand, ignore the challenge of learning in a continuous domain by discretizing actions [26, 45]. This discretization can introduce several drawbacks: It discards the inherent structure of the continuous space, increases complexity by adding a separate quantization step, and may limit expressiveness or accuracy when fine-grained control is required. Furthermore, the popular approach of learning a VQ-VAE [52] of the action space involves nondifferentiable operations and therefore requires advanced tricks during representation learning to sidestep optimization difficulties and instabilities [24, 20, 18, 31]. Hence, our goal is to design autoregressive transformers that preserve the continuous nature of the action space, leveraging the smooth and potentially multimodal behavior patterns that experts exhibit.

Diffusion policies, on the other hand, are adept at capturing continuous action distributions, but face two critical challenges in control tasks. First, real-time applications like robotics demand fast, efficient inference, yet diffusion models rely on iterative sampling, requiring dozens of forward passes

---

Correspondence: Ziyad Sheebaelhamd <ziyad.sheebaelhamd@uni-tuebingen.de>

39th Conference on Neural Information Processing Systems (NeurIPS 2025).

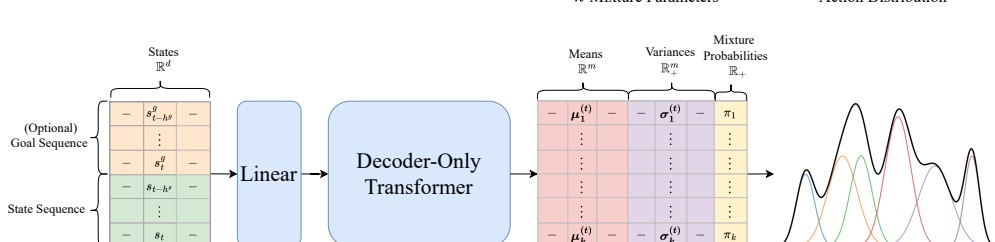

(a) An overview of the Q-FAT architecture.

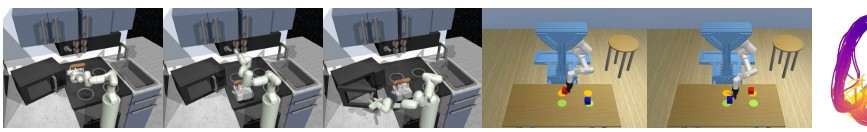

(b) Sample trajectories generated by a Q-FAT policy.

Figure 1: Figure (a) shows the overview of QFAT. A sequence of $h^s$ previous states and $h^g$ goal states are projected into the Transformer's embedding dimension using a linear layer. The Transformer then predicts the action distribution by predicting the GMM means, variances and mixture probabilities. Figure (b) shows a sample of generated trajectories from 3 representative environments, demonstrating the captured multi-modality in the sequence of solving tasks (Kitchen and UR3 Block Push) or the direction from which an object is approached (PushT).

to generate actions. This makes them orders-of-magnitude slower than autoregressive or feedforward policies [26]. Second, the downstream task of imitation learning is often reinforcement learning [43, 39, 58, 17], which typically requires exact action likelihoods for tasks like exploration and off-policy corrections [49, 1, 27, 56]. Diffusion models, however, generate samples by integrating (stochastic) differential equations [48, 47], making likelihood estimation computationally expensive.

To address these limitations, we propose the use of a Generative Infinite Vocabulary Transformer (GIVT) [50] that models the policy as a Gaussian Mixture Model (GMM) on top of a decoder-only transformer architecture [53, 41]. This approach enables modeling continuous actions as "tokens" by directly parameterizing the components of the GMM in each time step, avoiding the need for discretization and thus preserving the inherent geometry of the action space. This choice allows us to exploit the benefits of large-scale sequence modeling, such as autoregressive policy generation and access to explicit likelihood estimates, while avoiding the pitfalls of artificially segmenting continuous control signals into discrete bins. In addition, we discuss two different sampling strategies that reduce variability in the generated trajectories.

In summary, our work:

- Eliminates the need for action quantization in autoregressive policy parameterization, simplifying behavioral cloning pipelines while retaining the ability to directly evaluate action likelihoods.
- Proposes two sampling algorithms to reduce the variance of the generated trajectories, including a global mode-finding algorithm.
- Demonstrates state-of-the-art performance on standard benchmarks, both for conditional and unconditional policy generation using proprioceptive and image-based states, confirming the effectiveness of our method for multimodal policy modeling.

The implementation is available at `https://github.com/ziyadsheeba/qfat`.

## 2 Sequence to Action Prediction

We consider the following standard setting for imitation learning of simulated robotics tasks. An agent equipped with continuous controls $\mathcal{A}$ evolves in a continuous state space $\mathcal{S}$ in discrete time. In

behavioral cloning, we assume that a so-called *behavior policy* $\pi_b$ collected a dataset $D = \{\tau_i\}_{i=0}^{N-1}$ of $N$ trajectories in the environment, where each trajectory $\tau_i = \{s_0^i, a_0^i, \ldots, a_{n^i-1}^i, s_{n^i}^i\}$ is a sequence of states $s_t^i \in \mathcal{S}$ and actions $a_t^i \in \mathcal{A}$, with $n^i$ denoting the length of trajectory $i$. Note that we do not impose a Markovian assumption on the behavior policy. In general $a_t^i$ might depend on a history of states $\{s_j^i\}_{j=t-h}^t$, where $h$ is a fixed window.

Our objective is to learn a *sequence-to-action* predictor, or *imitating policy* $\pi_\theta$. We treat the objective as a supervised learning problem, where the parameter $\theta$ of our policy is optimized to minimize the negative log-likelihood with respect to $\pi_b$, $\mathbb{E}_{\tau \sim \pi_b}[-\log(\pi_\theta)(\tau)]$. To do so, we minimize:

$$\mathcal{L}(\theta) = -\frac{1}{N} \sum_{i=0}^{N-1} \sum_{t=0}^{n^i-1} \log \pi_\theta(a_t^i \mid s_{t-h:t}^i). \tag{1}$$

Note that a standard result in variational inference [15] shows that this objective is equivalent to minimizing the Kullback-Leibler (KL) divergence between the behavioral policy $\pi_b$ and the learned policy $\pi_\theta$, implying that the learned policy $\pi_\theta$ should match, or clone, the behavior of $\pi_b$. However, the choice of parametrization is crucial to guarantee a good approximation of the behavior policy.

## 3  The Q-FAT Algorithm

Current transformer-based policy parametrization approaches [26, 45, 32] consist of first discretizing the action space, e.g. using a VQ-VAE [52, 57], to comply with the standard transformer architecture requiring a discrete vocabulary. Inspired by the success of GIVT [50] in the image generation domain, our model *Quantization-Free Autoregressive Action Transformer* (Q-FAT) adapts GIVT to the continuous action behavioral cloning setting, modeling the action distribution at each step by *predicting the parameters of a multivariate Gaussian mixture model (GMM)*. Namely, Q-FAT models the probability of an action $a_t \in \mathbb{R}^m$ at step $t$ as

$$\pi_\theta\big(a_t \mid s_{t-1}, \ldots, s_{t-h^s}\big) = \sum_{i=1}^k \pi_i^t \, \mathcal{N}\big(a_t \mid \boldsymbol{\mu}_i^t, \boldsymbol{\sigma}_i^t\big), \tag{2}$$

where $\pi_i^t$, $\boldsymbol{\mu}_i^t$, and $\boldsymbol{\sigma}_i^t$ are produced by the transformer decoder [53, 41] at step $t$, conditioned on a history of previous states $\{s_i\}_{i=t-h^s}^{t-1}$ with $s_i \in \mathbb{R}^d$. Using a state history of length $h^s$ instead of only using the most recent state accounts for the potential non-Markovian structure in the demonstration dataset. Rather than predicting the full covariance $\boldsymbol{\Sigma}_i^t$ per mixture component, we assume a diagonal Gaussian distribution and only predict the diagonal entries of the covariance matrix $\boldsymbol{\sigma}_i^t$. While this simplifying assumption enables efficient likelihood computations, it does not impose independence across action dimensions in general. Q-FAT is trained using standard teacher forcing with a causal attention mask. We highlight that we only use state feedback and do not condition on the predicted actions. We discuss the important architecture decisions in Appendix D.

### 3.1  Goal Conditioning

For some tasks such as tracking, a near-future goal can be appended to the input to *condition* the resulting policy. The conditioning is formalized by defining a set of *goal states* $\mathcal{G} \subseteq \mathcal{S}$. We then learn a goal-conditioned policy $\pi_\theta\big(a_t \mid s_{t-1}, \ldots, s_{t-h^s}, g\big)$ where $g \subseteq \mathcal{G}$. In practice, we define $g$ as a sequence of future states to be reached, *i.e.*, $g = \{s_i\}_{i=t}^{t+h^g}$, where the goal states are typically generated from a high-level planner that plans instrumental goal states as done in Gupta et al. [11]. In our experiments, we simulate goal states from the demonstration dataset.

### 3.2  Action Sampling from a $k$-GMM

QFAT samples the next action from the conditional distribution, $a_t \sim \pi_\theta(\cdot|s_{t-1}, \ldots, s_{t-h^s}, g)$, which is a GMM with $k$ components. In practice, the model can be trained to output a short sequence of actions that are then executed in an open-loop fashion. This is done to ensure action consistency and avoid over-fitting to idle actions [7]. For simplicity, we describe only the next-action generation setting.

One can directly sample from the GMM that parametrizes the policy. For the *vanilla GMM sampling*, the procedure is:

1. Sample a mixture index $k$ from the categorical distribution defined by the probabilities $\{\pi_i\}_{i=1}^{k}$.
2. Sample an action $\mathbf{a}$ from the corresponding Gaussian component $\mathcal{N}(\boldsymbol{\mu}_i, \boldsymbol{\sigma}_i^2)$.

While this approach directly samples from the learned policy distribution, it can degrade performance in practice when the dataset is noisy. In that case, the GMM may overfit the data distribution and produce *noisy* actions, resulting in jitter or unstable trajectories. We discuss below how Q-FAT can be easily equipped with stabilizing sampling techniques that directly exploit the fitted GMM distribution.

### 3.3 Stabilizing Output Trajectories

When controlling real-world robotics systems, it is important to generate action sequences without jitter, as high frequency inputs may damage actuators and excite undesirable structural modes [2]. Thanks to the richness of the GMM representation of the policy, we can directly design stabilization patches.

**Down-scaling the Mixture Variances**    A common heuristic to mitigate noisy samples is to adjust the sampling temperature, which in our GMM parametrization is equivalent to *down-scaling the variance* in each Gaussian component [29, 16, 50, 51]. Concretely, one replaces each variance $\boldsymbol{\sigma}_i^2$ by $\alpha \boldsymbol{\sigma}_i^2$, for some small $\alpha \in (0, 1)$—sometimes by several orders of magnitude ($\alpha \ll 1$).

This compresses each component, forcing sampled actions to concentrate more tightly around each component's mean. However, merely shrinking the variances does not address the large-scale spread when mixture components are far apart. The inter-component variance remains, and noisy samples may still arise from small but non-negligible transitions among disparate components.

This phenomenon is exacerbated when the choice of the number of components $k$ is overspecified (see Figure 2 and further discussion in Appendix A). Furthermore, it has been shown by Carreira-Perpinan [5] that GMMs in dimension $d \geq 2$ may have a mode that does not directly align with the mean of any component. In such a situation, the down-scaling technique is likely to result in an insidious loss of the extra mode (see Figure 2).

**Mode-tracking via the Mean-Shift Algorithm** We explored a principled way to reduce the variance of sampling from multimodal distributions by *explicitly identifying the major modes* and then *sampling from those modes* with an appropriate probability while controlling variance. An effective way to find the modes in a GMM is via the mean-shift algorithm [5], which is an efficient fixed-point iteration algorithm (see Appendix B for further details). We propose a multinomial distribution over the modes[1] $\{m_1, ...m_J\}$ that respects the shape of the density:

$$\Pr(m_j) = \frac{w_j}{\sum_{j'} w_{j'}}, \tag{3}$$

$$\text{where } w_j = p_{\text{GMM}}(m_j) \cdot \left| -H_j \right|^{-\frac{1}{2}} \tag{4}$$

$$\text{and } H_j = \nabla^2 \log p_{\text{GMM}}(m_j). \tag{5}$$

Note that each $H_j$ admits a closed-form expression and can be computed in $O(km^2)$ operations. One can then further inject noise to the sampled mode using a Gaussian noise with a fixed variance or use the Hessian around the mode to account for the curvature of the distribution to determine the variance (see more details on this process in Appendix B). We emphasize that this mode-tracking algorithm can be efficiently implemented on GPUs using common deep learning libraries and introduces a minor overhead compared to the transformer forward pass.

## 4 Experiments

We evaluate the performance of Q-FAT through extensive experiments across 9 different tasks using five representative simulated robotics environments: PushT [7], Kitchen [11], UR3 BlockPush,

---

[1]The number of modes $J$ is hard to control theoretically [5].

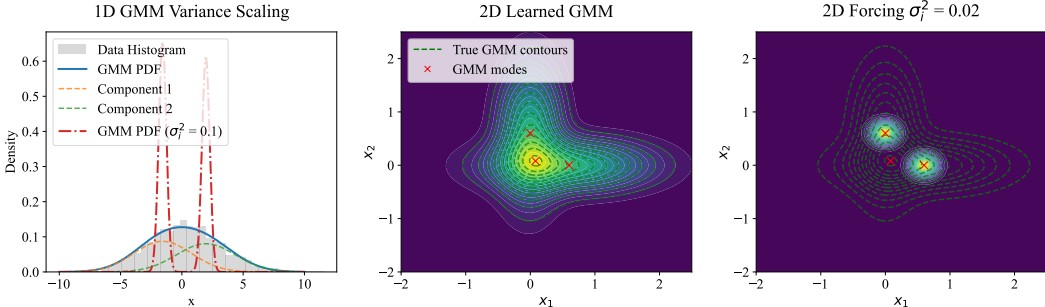

Figure 2: The figures demonstrate the potential detrimental effects of down-scaling the component variances in a learned Gaussian Mixture Model (GMM). The left figure illustrates how down-scaling variance introduces irreducible variance proportional to the distance between component means, when the number of components ($k$) is misspecified (e.g., approximating a unimodal distribution with two components). The two figures on the right show how reducing variance can lead to a loss of multimodality, specifically causing the central mode to disappear in a 2D two-mixture Gaussian that presents three modes.

BlockPush [10] and Multimodal Ant [26]. These environments cover the most common control signals in practice, namely position and velocity of robot's end effector and robot joint angles. A detailed description of each environment is provided in Appendix C. We further ran preliminary experiments on an autonomous driving dataset (nuScene [4]) and we add the experimental results in Appendix E. Our experiments are designed to address the following key research questions:

1. Is Q-FAT competitive with state-of-the-art methods in both conditional and unconditional behavior imitation?
2. Can Q-FAT effectively capture the diversity of demonstration datasets while avoiding mode collapse?
3. What is the effect of different sampling techniques on the performance of Q-FAT?
4. How sensitive is Q-FAT's performance to the choice of the mixture components $k$ in the learned policy?
5. How does the inference speed of Q-FAT compare to VQ-BeT?

We find that Q-FAT achieves state-of-the-art performance across the majority of conditional and unconditional tasks, achieving state-of-the-art balance between the quality and diversity of the generated actions. We report the results of Q-FAT using the variance reduced sampling (Section 3.2) with a factor of $10^{-6}$ and we discuss the implications of this choice in Section 4.3. Further training details are provided in Appendix D. A summary of our results is presented in Table 1.

## 4.1 Baselines and Performance Metrics

**Baselines** We compare Q-FAT against a range of state-of-the-art baselines. We restrict our focus to comparing against diffusion and transformer-based policy learning methods. For unconditional tasks, the baselines include: (1) BeT [45], which performs action discretization using k-means clustering and models the action distribution with a transformer decoder; (2) VQ-BeT [26], an extension of BeT that leverages vector quantization via a hierarchical Variational Autoencoder (VQ-VAE) [57] for action discretization; and (3) two variants of Diffusion Policy [7], namely a convolutional-based implementation and a transformer-based implementation. For goal-conditional tasks, we extend the comparison to include conditional versions of BeT and VQ-BeT, as well as two additional baselines: BESO [42], a denoising diffusion-based method with a conditional variant (C-BESO) and a classifier-free guided variant (CFG-BESO).

## 4.2 Quality and Diversity of Q-FAT's Actions

**Evaluation Metrics** We assess performance using *task success rate* and *induced behavioral entropy*. Task success is measured differently per environment: in Kitchen, Multimodal Ant and UR3 Block-

Table 1: Performance comparison between unconditional and conditional behaviors on the task success rates. We report the performance of Q-FAT with variance down-scaling with a factor of $10^{-6}$.

| — Unconditional Tasks — | | | | | | |
|---|---|---|---|---|---|---|
| Environment | Metric | BeT | DiffPolicy-C | DiffPolicy-T | VQ-BeT | Q-FAT |
| PushT | Final IoU ($\cdot/1$) | 0.39 | 0.73 | 0.74 | 0.78 | **0.80** |
| Image PushT | | 0.01 | 0.66 | 0.45 | 0.68 | **0.69** |
| Kitchen | Goals ($\cdot/4$) | 3.07 | 2.62 | 3.44 | 3.66 | **3.84** |
| Image Kitchen | | 2.48 | 3.11 | 3.01 | 2.98 | **3.15** |
| Multimodal Ant | | 2.73 | 3.12 | 2.90 | 3.22 | **3.30** |
| UR3 BlockPush | Goals ($\cdot/2$) | 1.59 | 1.83 | 1.82 | 1.84 | **1.99** |
| BlockPush | | 1.67 | 0.47 | **1.93** | 1.79 | 1.77 |
| — Conditional Tasks — | | | | | | |
| Environment | Metric | C-BeT | C-BESO | CFG-BESO | VQ-BeT | Q-FAT |
| Cond. Kitchen | Goals ($\cdot/4$) | 0.15 | 3.75 | 3.47 | **3.78** | **3.78** |
| Cond. UR3 BlockPush | Goals ($\cdot/2$) | 0.00 | 1.14 | 0.92 | 1.94 | **1.96** |

Figure 3: Overview of Q-FAT's behavioural entropy on unconditional behavior generation compared to baselines.

Push, the task success rate corresponds to the number of goals achieved per roll-out, while in PushT, task success is quantified using the *final* Intersection over Union (IoU) between the object's position and the target area. Behavioral diversity is measured by computing task completion sequence entropy (length 4 for Kitchen and Multimodal Ant, length 2 for UR3 BlockPush), reflecting variability in task completion. Baseline performance follows metrics from Lee et al. [26], and results are reported as the average success rate of the best checkpoint over 1000 episodes, including behavioral entropy where applicable.

**Comparison to Baselines** Q-FAT achieves state-of-the-art performance on the success metrics, as shown in Table 1. The performance gap observed with BeT can be attributed to the limitations of k-means clustering in high-dimensional spaces, which may hinder its ability to effectively capture complex action distributions. This is particularly exacerbated due to BeT attempting to discretize the whole unconditional action space, and not the conditional one. In the case of VQ-BeT, the sequential nature of first predicting a discrete action token followed by an action correction step introduces two potential sources of error: inaccuracies in token prediction and errors in the subsequent correction

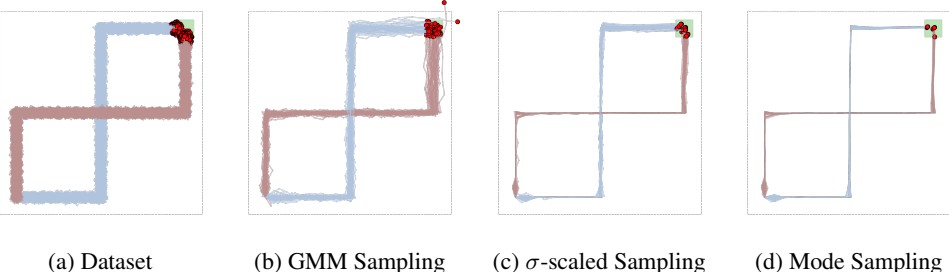

| (a) Dataset | (b) GMM Sampling | (c) $\sigma$-scaled Sampling | (d) Mode Sampling |
|---|---|---|---|

Figure 4: Sampling techniques from an 8-mixture Q-FAT policy on a multiroute environment [45]. (a) Raw dataset with two pairs of equally likely paths (blue and red) from start to target (green). (b) Direct GMM sampling yields noisy samples due to the captured dataset noise. (c) Variance scaling $(10^{-8})$reduces per-component variance but not the variance from inter-component distances.(d) Mode sampling largely suppresses noise while preserving dataset multimodality.

mechanism. This places significant reliance on the vector quantization network, which is challenging to optimize due to the need to discretize the entire action space and the required two-stage training process — first for the quantization network and then for the policy. For diffusion-based policies, the absence of an explicit likelihood function makes it difficult to control the variance of sampled actions, particularly in the presence of noisy data. This can be particularly problematic in fine-grained control tasks, where precise action execution is crucial for success. Consequently, while these approaches demonstrate strong performance in certain scenarios, they may struggle in tasks requiring high accuracy and reliability under noisy data. Q-FAT largely outperforms or matches the behavioral entropy of all baselines (Figure 3), which assesses diversity across the generated motion.

## 4.3 Effects of Sampling Techniques

We assessed the effect of the different sampling techniques discussed in Section 3.2. We found that down-scaling the component variances had a 7% and a 6.5% increase in the success performance metrics in the Kitchen environment and PushT, respectively, compared to the GMM sampling, while only seeing a marginal 2% drop in the diversity of the generated behaviors in the Kitchen environment.

The modes sampling matched the performance of the variance down-scaling on the evaluation metrics. This could be attributed to the fact that the learned mixture components are sufficiently distant in the environments we explored, thus the effects discussed in Section 3.2 are not pronounced.

Despite the quantitative metrics not showing significant differences between the variance down-scaling method and the mode sampling, we visualize the trajectories produced by both sampling techniques on a toy multi-route environment introduced by Shafiullah et al. [45] and see the effects of the irreducible variance in Figure 4. We further display a similar effect in the PushT environment in Figure 5.

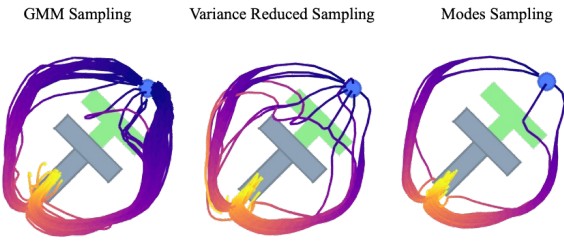

Figure 5: We visualize 400 trajectories generated from Q-FAT with 16 mixtures on the PushT environment with different sampling methods. One can see that mode sampling reduces the variance and does not produce trajectory artifacts.

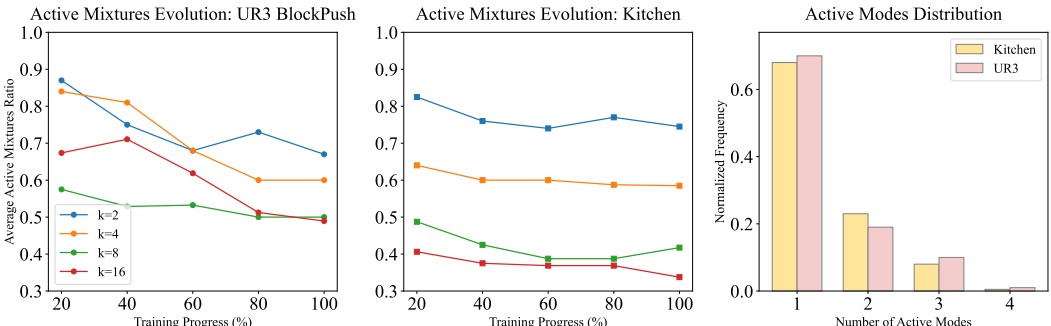

Figure 6: The left two plots display the evolution of the number of active mixtures during inference as training progresses, showing a clear pruning effect. On the rightmost plot, we display the distribution of the number of active modes at each timestep during inference for a converged model. One can see that trajectories remain largely unimodal.

### 4.4 Effect of the Number of Mixtures

We systematically evaluated the effect of varying the number of mixture components $k$ in Q-FAT, where $k$ controls the number of modes in the conditional action distribution for a given state context.

Specifically, we considered $k \in \{2, 4, 8, 16\}$ in two benchmark domains, the Kitchen and UR3 BlockPush environments. Our results indicate that Q-FAT's success performance is highly robust across these values of $k$, with negligible degradation in performance. In the Kitchen environment, we observed an 18% increase in behavioral entropy when $k$ was increased from 2 to 4, after which entropy remained approximately constant as $k$ continued to increase. In contrast, in the UR3 BlockPush environment, the behavioral entropy remained stable regardless of $k$. The key to achieving this robustness is careful initialization.

In particular, we found it important to initialize each mixture component's mean to be distant from the others. Concretely, we placed these means on a hyper-cube spanning the range $[-1, 1]$ per dimension, maximizing their mutual separation, and assigned each component a unit variance. The range $[-1, 1]$ is a reasonable choice, as actions are typically bounded in practice and normalized in our experiments.

To investigate how the mixture components were utilized during inference, we tracked the average number of *active* components over the course of training (see Figure 6, left). We observed that the model effectively prunes redundant mixture components as training progresses, with the pruning effect becoming more pronounced when more components are initially employed. Interestingly, in the Kitchen environment, which has a nine dimensional action space, less pruning occurs as $k$ increases compared to the UR3 environment, whose action space is only two dimensional. A plausible explanation for this discrepancy lies in the dimensionality of the action space.

In higher dimensions, each Gaussian component covers a proportionally smaller volume of the action space. Consequently, the model requires more components to adequately represent the conditional action distribution, and therefore prunes fewer mixture components. In lower-dimensional environments, by contrast, each component can cover a larger fraction of the data, which makes aggressive pruning more viable.

We further plotted the distribution over the number of active modes for the Kitchen and the UR3 BlockPush environments. We observed that approximately 70% of the course of a trajectory is uni-modal (Figure 6, right). This shows that multi-modality is required only a minority of the time during a rollout. This observation could be useful for future work to exploit this uni-modality computationally.

### 4.5 Inference Time Analysis

To compare computational efficiency, we evaluated the inference times of Q-FAT and VQ-BeT on a 16 GB MacBook Pro CPU. The analysis utilized models trained in the Kitchen environment, with hyperparameters for Q-FAT specified in Table 3 and for VQ-BeT as described in [26]. Q-FAT

is approximately 2x faster on the tested hardware due to its efficient action decoding head. The breakdown of inference time per model component is detailed in Table 2.

Table 2: Inference time breakdown for Q-FAT and VQ-BeT on the models used for the Kitchen environment. Q-FAT has a lighter action-decoding head, resulting in a signigicant speedup.

| Component | VQ-BeT (ms) | Q-FAT (ms) |
|---|---|---|
| Shared Backbone (minGPT) | $\sim$1.0 | $\sim$1.0 |
| Action Decoding Head | $\sim$1.2 | $\sim$0.1 |
| **Total Inference Time** | $\sim$**2.2** | $\sim$**1.1** |

The primary performance difference arises from the action decoding head. While both models use an identical backbone architecture, VQ-BeT's reliance on two separate 2-layer MLPs for action decoding creates a computational bottleneck. In contrast, Q-FAT's use of a single, efficient linear action decoding layer results in a significant speedup.

## 5   Related Work

**Offline Learning for Decision Making**   Learning to act from offline data has received significant attention over the years due to its vast potential in practical applications. This line of research can be divided into two categories: offline reinforcement learning [25, 27, 22, 6] and imitation learning [34, 37, 38, 13]. In offline reinforcement learning, on the one hand, the goal is to learn a policy from demonstrations with suboptimal behaviors, which necessitates the access to rewards in the demonstration dataset. Imitation learning, on the other hand, assumes that the trajectories are collected from near optimal behaviors, avoiding access to reward signals in the collected datasets. This increases the attractiveness of imitation learning in practical setups, since reward signals are hard to define [9, 23]. Q-FAT falls under the framework of behavioral cloning, which is a form of imitation learning where the policy learning is treated as a supervised learning problem.

**Generative Models for Cloning Behavior**   Complex generative models have been employed to capture the full data diversity due to the complex and multimodal behavior in human and robot datasets. Earlier work has explored the use of Gaussian processes [46], energy-based models [10] and generative adversarial networks [21] as policy parametrizations. More recently, more attention has been directed towards more powerful models, such as transformers [45, 26, 32, 55] and diffusion models [7, 54, 40, 36]. Unlike other transformer-based policies, Q-FAT uses a continuous parametrization of the output by predicting a GMM. Note that this idea has been explored before using LSTM backbones [29, 45], however, it has been shown to produce poor performance [45, 26], which we attribute to the instability of training LSTM networks and their limited capacity to model long sequences [14].

## 6   Discussion and Future Work

In this work, we introduced Q-FAT, a quantization-free action transformer that achieves state-of-the-art performance across various simulated robotics environments, opening new research directions in both behavioral modeling and reinforcement learning. Compared to prior work [26], Q-FAT overcomes the challenging non-differentiable step in action discretization while maintaining or improving performance.

By integrating Q-FAT into end-to-end policy learning pipelines, future research could evaluate its influence on sample efficiency and final task performance. Another promising area involves incorporating Bayesian priors into Q-FAT's action distribution estimates, which could facilitate active exploration in complex, high-dimensional settings. For fine manipulation tasks, researchers could further investigate coarse-to-fine sampling strategies based on Gaussian mixture model representations, potentially improving both exploration breadth and control. Finally, extending Q-FAT to non-Euclidean action spaces, such as those with Riemannian geometry, may enable more accurate and natural representations for tasks like legged locomotion or dexterous manipulation.

# 7 Limitations

While we have demonstrated that Q-FAT is effective in learning complex action distributions, the underlying GMM loss function used to train the policy assumes a Euclidean action space. In some environments (e.g., humanoid robots), the action space has a Riemannian geometry, which could potentially result in difficulty during learning. This can be counteracted by mapping actions into a latent Euclidean space using an action autoencoder [19] and perform the GMM loss in the latent space. However, unlike previous work that uses discrete action encoders [26, 45], Q-FAT allows the flexibility of using continuous latent spaces, allowing end-to-end differentiability of the joint encoder and policy models. While this is an exciting research direction, we leave this for future work.

# 8 Impact Statement

Our work tackles simulated continuous control tasks whose downstream consequences are important for robotics. The probabilistic nature of our method may impact the safety of the generated behavior and should be further studied and tested. However, unlike prior work, our approach enables uncertainty estimates directly in the action space, which have the potential to improve safety by providing uncertainty quantification.

# 9 Acknowledgment

We would like to thank Youssef Saied for the valuable discussions throughout the project and Nicolas Nguyen for reviewing the manuscript. C.V. and Z.S. are funded by the German Research Foundation (DFG) under both the project 468806714 of the Emmy Noether Programme and under Germany's Excellence Strategy – EXC number 2064/1 – Project number 390727645. M.M. is funded by the German Research Foundation (DFG) under the project 456587626 of the Emmy Noether Programme. We also thank the international Max Planck Research School for Intelligent Systems (IMPRS-IS) for the support.

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

## A  GMM Properties

**GMM Moments**

Let $\mathbf{X}$ be a random variable from a $k$-mixture GMM distribution in $\mathbb{R}^d$. The probability density function can then be written as:

$$p(\mathbf{x}) = \sum_{i=1}^{k} \pi_i \mathcal{N}(\mathbf{x} \mid \boldsymbol{\mu}_i, \boldsymbol{\Sigma}_i), \tag{6}$$

where $\mathcal{N}(\mathbf{x} \mid \boldsymbol{\mu}_i, \boldsymbol{\Sigma}_i)$ is the multivariate Gaussian distribution for the $i$-th component. The mean and the covariance of $\mathbf{X}$ are given by:

$$\mathbb{E}[\mathbf{X}] = \sum_{i=1}^{k} \pi_i \boldsymbol{\mu}_i, \tag{7}$$

$$\mathrm{Cov}[\mathbf{X}] = \sum_{i=1}^{k} \pi_i \left( \boldsymbol{\Sigma}_i + (\boldsymbol{\mu}_i - \mathbb{E}[\mathbf{X}])(\boldsymbol{\mu}_i - \mathbb{E}[\mathbf{X}])^\top \right), \tag{8}$$

where $\pi_i$ are the mixing coefficients, and $\boldsymbol{\mu}_i$, $\boldsymbol{\Sigma}_i$ are the mean and covariance parameters for the $i$-th Gaussian component. Note that while the mean of the GMM is a weighted sum of the component means, the covariance is not. The total covariance reflects both the inherent variability within each component and the variability introduced by the differences in component means.

## B  GMM Mode Finding and Sampling

**Mode Finding**

To find the modes of the GMM, we compute the gradient of the probability density function with respect to $\mathbf{x}$ and equate it to zero:

$$\nabla p(\mathbf{x}) = \sum_{i=1}^{k} \pi_i \nabla \mathcal{N}(\mathbf{x} \mid \boldsymbol{\mu}_i, \boldsymbol{\Sigma}_i) = \mathbf{0}, \tag{9}$$

where the gradient of the multivariate Gaussian distribution is given by

$$\nabla \mathcal{N}(\mathbf{x} \mid \boldsymbol{\mu}_i, \boldsymbol{\Sigma}_i) = \mathcal{N}(\mathbf{x} \mid \boldsymbol{\mu}_i, \boldsymbol{\Sigma}_i) \boldsymbol{\Sigma}_i^{-1}(\boldsymbol{\mu}_i - \mathbf{x}). \tag{10}$$

We obtain therefore:

$$\sum_{i=1}^{k} \pi_i \mathcal{N}(\mathbf{x} \mid \boldsymbol{\mu}_i, \boldsymbol{\Sigma}_i) \boldsymbol{\Sigma}_i^{-1}(\boldsymbol{\mu}_i - \mathbf{x}) = \mathbf{0}. \tag{11}$$

We identify the modes of the GMM by computing all critical points $\mathbf{x}^*$ that satisfy (11). Moreover, to ensure that a critical point $\mathbf{x}^*$ is indeed a mode, the Hessian of the density function at that point must be negative definite. The Hessian of the GMM density is given by

$$\nabla^2 p(\mathbf{x}) = \sum_{i=1}^{k} \pi_i \nabla^2 \mathcal{N}(\mathbf{x} \mid \boldsymbol{\mu}_i, \boldsymbol{\Sigma}_i), \tag{12}$$

where the Hessian of the multivariate Gaussian distribution is:

$$\nabla^2 \mathcal{N}(\mathbf{x} \mid \boldsymbol{\mu}_i, \boldsymbol{\Sigma}_i) = \mathcal{N}(\mathbf{x} \mid \boldsymbol{\mu}_i, \boldsymbol{\Sigma}_i) \left[ \boldsymbol{\Sigma}_i^{-1}(\boldsymbol{\mu}_i - \mathbf{x})(\boldsymbol{\mu}_i - \mathbf{x})^\top \boldsymbol{\Sigma}_i^{-1} - \boldsymbol{\Sigma}_i^{-1} \right]. \tag{13}$$

Therefore, the Hessian of the GMM density becomes

$$\nabla^2 p(\mathbf{x}) = \sum_{i=1}^{k} \pi_i \mathcal{N}(\mathbf{x} \mid \boldsymbol{\mu}_i, \boldsymbol{\Sigma}_i) \left[ \boldsymbol{\Sigma}_i^{-1}(\boldsymbol{\mu}_i - \mathbf{x})(\boldsymbol{\mu}_i - \mathbf{x})^\top \boldsymbol{\Sigma}_i^{-1} - \boldsymbol{\Sigma}_i^{-1} \right], \tag{14}$$

and a critical point $\mathbf{x}^*$ is a mode if: $\nabla p(\mathbf{x}^*) = \mathbf{0}$ and $\nabla^2 p(\mathbf{x}^*)$ is negative definite.

While GMMs are often parameterized by a finite number of components, the resulting distribution can exhibit a richer structure, particularly in higher-dimensional spaces. Specifically, **in dimensions greater than two ($d > 2$), a GMM can possess more modes (local maxima) than the number of its constituent components** [5]. This phenomenon arises due to the interplay between the covariance structures and the relative positions of the Gaussian components. In higher dimensions, the overlapping regions of different components can create multiple peaks in the probability density function that cannot be attributed to individual components.

**Mean-Shift**

We start by solving (11) for $\mathbf{x}$ as follows:

$$\mathbf{x} \;=\; \left(\sum_{i=1}^{k} \pi_i\, \mathcal{N}\!\left(\mathbf{x} \,\middle|\, \boldsymbol{\mu}_i, \boldsymbol{\Sigma}_i\right) \boldsymbol{\Sigma}_i^{-1}\right)^{-1} \sum_{i=1}^{k} \pi_i\, \mathcal{N}\!\left(\mathbf{x} \,\middle|\, \boldsymbol{\mu}_i, \boldsymbol{\Sigma}_i\right) \boldsymbol{\Sigma}_i^{-1}\, \boldsymbol{\mu}_i\,. \tag{15}$$

To obtain the mean-shift update in practice, one can treat the solution to (15) as a *fixed-point iteration*. More precisely, we define the operator $T : \mathbb{R}^d \rightarrow \mathbb{R}^d$ by

$$T(\mathbf{x}) = \left(\sum_{i=1}^{k} \pi_i\, \mathcal{N}\!\left(\mathbf{x} \,\middle|\, \boldsymbol{\mu}_i, \boldsymbol{\Sigma}_i\right) \boldsymbol{\Sigma}_i^{-1}\right)^{-1} \sum_{i=1}^{k} \pi_i\, \mathcal{N}\!\left(\mathbf{x} \,\middle|\, \boldsymbol{\mu}_i, \boldsymbol{\Sigma}_i\right) \boldsymbol{\Sigma}_i^{-1}\, \boldsymbol{\mu}_i\,. \tag{16}$$

Then, starting from an initial guess $\mathbf{x}^{(0)}$, the mean-shift procedure updates the estimate of $\mathbf{x}$ via

$$\mathbf{x}^{(t+1)} \;=\; T\!\left(\mathbf{x}^{(t)}\right).$$

The fixed-point iteration has the following interpretation: each iteration *shifts* the current iterate after $t$ iterations closer to the modes (high-density regions) of the underlying density defined by the mixture of Gaussians. In practice, it has been shown that initializing $\mathbf{x}^{(0)}$ with the component means $\boldsymbol{\mu}_i$ results in fast convergence [5], which we also observe in our experiments. To account for the fact that the number of modes could exceed the number mixture components, we add extra initialization points from the convex-hull of the component centroids capturing the most important modes.

**Mean-Shift with Diagonal-Covariance Components**

An important special case arises when the covariance matrices $\boldsymbol{\Sigma}_i$ are *diagonal*. In that scenario, let

$$\boldsymbol{\Sigma}_i \;=\; \mathrm{diag}\!\left(\sigma_{i,1}^2, \ldots, \sigma_{i,d}^2\right).$$

Since each $\boldsymbol{\Sigma}_i^{-1}$ is also diagonal (with entries $1/\sigma_{i,j}^2$ along the diagonal), the vector and matrix operations inside $T(\mathbf{x})$ decouple across dimensions. Hence the inverse of $\sum_{i=1}^{k} \pi_i\, \mathcal{N}(\mathbf{x} \mid \boldsymbol{\mu}_i, \boldsymbol{\Sigma}_i)\, \boldsymbol{\Sigma}_i^{-1}$ reduces to a diagonal matrix whose components can be computed efficiently. As a result, each coordinate of the updated point can be determined independently, making the mean-shift iteration particularly fast on modern hardware.

**Modes Sampling**

To sample the modes proportional to their coverage of the data support, we approximate the integral of the GMM density in the local neighborhood of each mode using a *Laplace approximation* and use the value of the integral around each mode to compute the relative weight of the mode. We summarize the full mode-finding and sampling procedure in Algorithm 1.

## C  Environment Details

In our experiments, we evaluated Q-FAT in three distinct environments: Kitchen, PushT, and UR3 BlockPush. In the following, we provide a brief description of each environment and its variants.

- **Franka Kitchen**: This environment involves a Franka Panda robotic arm with a nine dimensional action space, designed for multi-task manipulation in a simulated kitchen

---

**Algorithm 1** Mode Extraction and Sampling

---

1: **Inputs:** GMM parameters $\{\pi_i, \boldsymbol{\mu_i}, \boldsymbol{\Sigma_i}\}_{i=1}^k$; tolerance $\epsilon$; rejection threshold $\theta$; max iterations max_it; number of initializations n_init;

2: **Initialize Mode Set:** $\mathcal{M} \leftarrow \emptyset$

3: **Initialize Seed Set:**

4: $\mathcal{X} \leftarrow \text{Uniform\_Sample}\big(\text{Convex\_Hull}(\{\boldsymbol{\mu}_i\}), \text{n\_init}\big)$

5: **for** $x \in \mathcal{X}$ **do**

6:     Iteration counter $i \leftarrow 0$

7:     Define the operator $T(.)$ as in Equation 16

8:     **repeat**

9:         Update $\mathbf{x}$ via fixed-point iteration: $\mathbf{x} \leftarrow T(\mathbf{x})$

10:        Increment $i$

11:    **until** $i \geq \text{max\_it}$ or $\|\mathbf{x} - \mathbf{x}_{\text{old}}\| < \epsilon$

12:    Compute $\mathbf{H} = \nabla^2 p(\mathbf{x})$

13:    **if** $\max(\text{eigenvalues}(\mathbf{H})) < 0$ **then**

14:        Merge nearby modes within a distance threshold, keeping the highest-probability mode

15:        Add mode to $\mathcal{M}$

16:    **end if**

17: **end for**

18:

19: **Compute Mode Weights:**

20: $w_j = p(\mathbf{x}_j)\big|-\nabla^2 \log p(\mathbf{x}_j)\big|^{-1/2}$, $j = 0, \ldots, |\mathcal{M}|$

21: **Sample mode:** Sample $j \sim \text{Categorical}\left(\{w_j\}_{j=0}^{|\mathcal{M}|}\right)$

22: **Return:** $\mathbf{x}_j$

---

setting [11]. The dataset consists of 566 human-collected demonstrations, where each trajectory completes a subset of four out of seven possible tasks in varying orders. For the image-based version of the environment, we rendered the collected trajectories into 112x112 images.

- **PushT**: The PushT environment focuses on pushing a T-shaped block to a target position on a table using two dimensional end-effector velocity control [7]. The dataset includes 206 human-collected demonstrations.

- **UR3 BlockPush**: In this task, a UR3 robotic arm is tasked with moving two blocks to designated goal circles on a table [10]. The dataset exhibits multimodality, as the blocks can be moved in either order. In the unconditional setting, we evaluate whether both blocks reach their respective goals. In the conditional setting, the model is provided with the target positions of the blocks, and performance is assessed based on the order in which the blocks reach their goals.

- **BlockPush**: The goal of this task is moving two (red and blue) blocks two targets [10]. The blocks can be moved in either order and can be put into either of the targets. While this environment is similar to UR3 BlockPush, it exhibits more stochasticity due to the random initialization of the block positions, and the targets being randomly translated and rotated. The dataset contained 1000 demonstrations.

- **Multimodal Ant**: The Multimodal Ant environment [26] is a modification of the Ant environment introduced by Schulman et al. [44] where the goal of the ant is to visit four distinct locations placed on vertices of a square. The multimodality in the dataset arises from the different order the goals are visited. The dataset contains 600 human-collected trajectories

# D  Training Details

For our experiments, we used the minGPT [3] backbone as the decoder-only transformer implementation. The hyperparameters used for each of the environments are detailed in Table 3.

## D.1  Architectural Design Choices

The policy was trained using teacher forcing with a causal attention mask, as this provided a stronger learning signal compared to computing the loss only on the last action given a context of states. We found that not using teacher forcing and employing a full attention mask significantly degraded performance. We further experimented with feeding back the sampled actions into the model's input by concatenating states and actions into a single feature vector. This approach degraded policy performance in simulated environments, likely due to the model overfitting to the highly correlated previous actions in the context while ignoring state information. Injecting Gaussian noise into the previous actions was necessary to achieve reasonable performance in this setting; however, this provided no discernible benefit compared to a policy without action feedback in our experiments.

## D.2  History Masking

During training, we observed instances where both validation and training losses decreased, yet the environment rewards declined. We attribute this phenomenon to causal confusion [8], where the model overfits the recent sequence of states. Instead of learning true causal relationships, the model relies on temporally correlated but non-causal patterns to predict future actions, leading to suboptimal performance. To mitigate this, we introduced history masking, randomly masking the context (excluding the current state) with a certain probability during training. Applying this technique in the Kitchen, Multimodal Ant, and BlockPush environments (with masking probabilities of 0.7, 0.3, and 0.3, respectively) led to an improved correlation between environment rewards and validation loss.

## D.3  Image-based Environments

For Image PushT and Image Kitchen, we fine-tuned a ResNet18 encoder [12], extracting feature maps from the first four layers to construct low-level features [35]. These features were then spatially average-pooled and fed into the transformer with a dimensionality of 1024. For Image PushT, we applied data augmentation (color jitter, random cropping, and random gray-scaling with probability 0.5), which proved crucial for preserving the invariance of the pre-trained ResNet and preventing overfitting to our dataset. Conversely, for Image Kitchen, we found data augmentation deteriorated performance.

## D.4  Goal Conditioning

In goal-conditional tasks, we conditioned the model on a sequence of future states equal in length to the state history. During inference, we randomly sampled a reference trajectory from the dataset. The agent's reward was computed only if it achieved the goal sequence in the same order as the reference trajectory.

## D.5  Evaluation

All training datasets were normalized using min-max scaling to ensure state and action features lie within the range $[-1, 1]$. Following Lee et al. [26], Shafiullah et al. [45], we split the data into 95% for training and 5% for validation. The policy was evaluated periodically during training using a small number of environment roll-outs (20 to 50). The models achieving the lowest validation loss and highest success metrics were selected for reporting.

## D.6  Model Size

For a fair comparison with VQ-BeT [26], we ensured our models have a comparable (or smaller) number of trainable parameters, on the order of $10^5 - 10^6$. We slightly deviated from their transformer encoder hyperparameters to account for the additional capacity utilized by their hierarchical vector

| Hyperparameter | PushT | Image PushT | Kitchen | UR3 | Multimodal Ant | BlockPush | nuScenes |
|---|---|---|---|---|---|---|---|
| Layers | 6 | 6 | 6 | 6 | 6 | 4 | 6 |
| Attention heads | 8 | 8 | 8 | 8 | 8 | 4 | 8 |
| Embedding dimension | 128 | 128 | 128 | 128 | 128 | 72 | 128 |
| Dropout probability | 0.1 | 0.1 | 0.1 | 0.1 | 0.1 | 0.1 | - |
| State history size | 5 | 5 | 10 | 10 | 10 | 3 | - |
| Action horizon | 5 | 5 | 1 | 1 | 1 | 1 | 6 |
| Training epochs | 1000 | 1000 | 1200 | 400 | 1000 | 1200 | 1000 |
| Batch size | 256 | 256 | 128 | 2048 | 128 | 256 | 128 |
| Number of mixtures $k$ | 4 | 4 | 4 | 4 | 4 | 2 | 2 |
| Maximum learning rate | 1e-3 | 1e-3 | 1e-3 | 1e-4 | 1e-3 | 1e-4 | 1e-4 |
| Minimum learning rate | - | - | 1e-6 | 1e-6 | 1e-6 | 1e-7 | - |
| Learning Rate Schedule | - | - | Cosine | Cosine | Cosine | Cosine | - |
| Optimizer | Adam | Adam | Adam | Adam | Adam | Adam | Adam |

Table 3: Environment hyperparameters.

quantization autoencoder for action discretization. Training hyperparameters, such as batch size and learning rates, were adjusted to accommodate the differences in loss functions between the methods. Further details are provided in Table 3 and Table 13 in Lee et al. [26].

### D.7 Hardware

Experiments were conducted on a heterogeneous cluster, making precise hardware control infeasible. However, all experiments were run on a single desktop-grade GPU with at most 32 GB of memory. Training a single model typically took 4-8 hours, depending on dataset size and the frequency of environment evaluations for validation.

## E nuScenes Experiment

To evaluate our model's applicability beyond robotic manipulation, we use the nuScenes [4] dataset, a large-scale benchmark for autonomous driving. We utilize the object-centric, preprocessed version of the dataset from Mao et al. [30], and follow the exact preprocessing and tokenization steps for driving data as detailed in the VQ-BeT paper [26]. The model's input is a set of tokens representing the driving mission (e.g., turn left), the ego-vehicle's current state (velocity, acceleration), its recent trajectory history, and the state of surrounding objects (position, class). The task is to predict the ego-vehicle's trajectory over the next six timesteps. Success is evaluated using two primary metrics: the average L2 distance between the predicted and ground-truth trajectories to measure accuracy, and the collision rate to assess the safety of the generated path.

For this sequential prediction task, we found it necessary to adapt our model's architecture. A naive approach of predicting a single high-dimensional vector concatenating all future waypoints resulted in degraded performance. This is likely because the model is forced to learn a direct mapping to a highly complex and multimodal joint distribution of future states. Such a method struggles to enforce temporal consistency, often producing kinematically implausible trajectories.

Table 4: Preliminary results from our initial experiments on the nuScenes autonomous driving benchmark. Lower values are better.

| Method | L2 Error (m) | Collision Rate (%) |
|---|---|---|
| Diffusion | 0.96 | 0.44 |
| VQ-BeT | **0.73** | **0.29** |
| Q-FAT (ours) | 0.75 | 0.37 |

Consequently, we adopted an autoregressive prediction scheme, where the model forecasts one waypoint at a time, conditioned on its previous predictions. This approach simplifies the learning problem by factorizing the joint distribution into a sequence of more manageable conditional distributions. By doing so, the model implicitly learns the transition dynamics of the environment, ensuring that the generated trajectory is temporally coherent and physically plausible. Furthermore, instead of predicting the absolute coordinates of each waypoint, our model predicts the *delta*, or displacement,

from the previous waypoint. This encourages the model to learn a policy that is invariant to the absolute frame of reference, which has been shown to improve generalization.

The results from these initial experiments are presented in Table 4. These findings show that Q-FAT achieves performance competitive with the VQ-BeT baseline, demonstrating that our method is effective in this challenging domain. The training hyperparameters can be found in Table 3.

