# OpenReview forum: "Quantization-Free Autoregressive Action Transformer"
_NeurIPS.cc/2025/Conference — NeurIPS 2025 spotlight_

### Official Review · Reviewer_Bc6V · 2025-06-26

**Clarity:** 3
**Significance:** 2
**Originality:** 2
**Rating:** 4
**Confidence:** 4

**Summary:**

The paper proposed that Quantization-Free Autoregressive Action Transformer (Q-FAT) replaces the discrete-token step of prior behavior models (e.g., BeT, VQ-BeT) with a decoder-only transformer that directly regresses continuous actions by outputting the means, variances, and logits of a Gaussian-mixture model at every time step. The paper claimed this could preserve the geometry of the action space and remove a separate VQ-VAE stage. To produce smooth yet multimodal trajectories, the authors introduce two generic post-processing techniques: temperature-based variance scaling and a mode-tracking mean shift. Experiments on nine tasks from five simulated‑robotics suites show higher success rates and comparable or better behavioural‐entropy diversity than diffusion‑policy and discrete‑token transformer baselines.

**Questions:**

1. What is the diversity and quality of the trajectories used for training? (I assume they are all expert-level.)

2. It would be beneficial to include some discussions comparing VQ-based approaches in the context of offline learning for decision-making (e.g., https://openreview.net/forum?id=cA77NrVEuqn, https://openreview.net/forum?id=pQsllTesiE, https://openreview.net/forum?id=n9lew97SAn).

**Ethical Concerns:**

["NO or VERY MINOR ethics concerns only"]

**Final Justification:**

The proposed method achieves strong experimental performance by effectively maintaining granularity in continuous action spaces. Additionally, as highlighted by the authors during discussions, the method demonstrates clear potential for extension to high-dimensional action spaces. Evaluating the performance of Q-FAT on more complex distributions in high-dimensional settings would be a valuable direction for future work.

**Limitations:**

Yes.

**Quality:**

3

**Strengths And Weaknesses:**

Strengths:
1. The authors conduct experiments across nine tasks, including proprioceptive and image-based environments, both conditional and unconditional, each with 1000 episodes.
2. Ablation studies are performed regarding the effects of sampling and mixture count.
3. By omitting the VQ-VAE quantization step used in BeT/VQ-BeT, the proposed method preserves the geometry of continuous controls and simplifies the behavioral cloning pipeline. Empirically, it demonstrates better performance compared to VQ-BeT, which includes the VQ step. Additionally, the GMM parameterization serves as a principled continuous extension of discrete tokens and provides analytic likelihoods, an advantage absent in diffusion policies.

Weaknesses:
1. The concept of removing the VQ step to maintain the granularity of the action space is not novel (e.g., https://openreview.net/forum?id=btpgDo4u4j). The core transformer approach originates from GIVT, and sampling techniques such as variance scaling are already established in diffusion models. Nevertheless, the presented method does demonstrate performance improvements over prior work, such as VQ-BeT, highlighting the effectiveness of increased granularity in learned actions.

2. While the VQ step indeed introduces additional training costs, it offers benefits in terms of learning robust representations, particularly when the dimensionality of the action space is high. Increased action-space dimensionality may arise in two scenarios: (1) when the action space itself is inherently large, or (2) when learning meaningful action chunks or temporal abstractions becomes critical. To substantiate the claim in the paper (that VQ-based methods limit expressiveness or accuracy in fine-grained control scenarios), a more detailed comparison between methods with VQ (e.g., VQ-BeT) and without VQ (the proposed method) in the described experimental settings would be beneficial.

---

> ### Author Rebuttal · Authors · 2025-07-31
>
> Thanks for your thorough evaluation and insightful feedback on our work. We will expand the discussion in our paper to include VQ-based approaches in the context of offline learning for decision-making. We address your questions and concerns below:
>
> - **The diversity and quality of the trajectories used for training**: In this work, we are in a behavioral cloning setup (Section 2) where there is no reward signal. This is consistent with the data used with the baseline methods used [Chi et. al 2023, Lee et. al 2024]. Thus, it is standard to assume that the demonstrations are expert level.
>
> - **Vector quantization offers benefits in terms of learning robust representations**: We agree that VQ-based methods could be effective, and our work does not dismiss that. Our central argument is that the VQ step breaks the continuous structure of the action space, which can limit expressiveness, and introduces significant complexity and optimization challenges. These include the need for advanced tricks to handle non-differentiable operations and instabilities during training. Furthermore, it complicates the pipeline by requiring a separate, two-stage training process first for the VQ-VAE and then for the policy. Q-FAT eliminates these issues entirely. We explained an argument why Q-FAT can be more effective (Section 4.1, lines 186-191) than VQ-based methods for fine-grained control. As for GMMs, they universally approximate any distribution given enough mixture components [Palataniotis et al. 2017]. Furthermore, recent work [Tschannen et. al 2024] has demonstrated scaling GMMs to high dimensional image modeling with an autoregressive transformer (in a 128-dimensional latent space with 1k mixture components), demonstrating high sample fidelity. This indicates that scaling GMMs to complex distributions in high-dimensional spaces, such as action spaces, is straightforward.
>
> ---
> ### References
> [Chi et. al 2023] Diffusion Policy: Visuomotor Policy Learning via Action Diffusion
> [Lee et. al 2024] Behavior Generation with Latent Actions
> [Palataniotis et al. 2017] Gaussian Mixtures and Their Applications to Signal Processing
> [Tschannen et. al 2024] JetFormer: An Autoregressive Generative Model of Raw Images and Text

---

> > ### Comment · Reviewer_Bc6V · 2025-08-01
> > **Official Comment by Reviewer Bc6V**
> >
> > Thank you for the response. It would be interesting to see the performance of Q-FAT on complex distributions in high-dimensional spaces in future work.

---

### Official Review · Reviewer_DqgJ · 2025-06-29

**Clarity:** 3
**Significance:** 3
**Originality:** 2
**Rating:** 4
**Confidence:** 3

**Summary:**

This paper adopts the GIVT model to represent the control policy of robotic systems. By using a Gaussian Mixture Model (GMM) as the output distribution, GIVT avoids the need for discretizing actions. The authors also propose a mode-tracking sampling technique to reduce variance. Experiments across five robotic environments demonstrate that Q-FAT achieves state-of-the-art performance.

**Questions:**

•	Real-world robotics and autonomous driving experiments are missing. While real robotic experiments might be difficult to conduct at this stage, it is important to include at least some experiments in autonomous driving scenarios (e.g., CARLA, nuScenes) to further demonstrate the practical utility of the proposed method.
•	The mode-finding process becomes increasingly expensive as the action dimensionality grows. It would be valuable to include an experiment evaluating how Q-FAT performs in such high-dimensional settings.
•	I couldn’t find details on how the demonstration datasets were collected in the appendix. Please indicate whether they were obtained from human demonstrations.

**Ethical Concerns:**

["NO or VERY MINOR ethics concerns only"]

**Final Justification:**

After reviewing the author's recent responses, I'm willing to increase my score.

**Limitations:**

Will Q-FAT still perform well when applied to experiments with inadequate demonstration data? Providing additional experiments or commentary on suboptimal performance would strengthen the paper.

**Quality:**

3

**Strengths And Weaknesses:**

Strengths:
•	The elimination of action quantization in transformer-based policies is an important problem in robotic control systems.
•	The author clearly explains the motivation for adopting GIVT and the Mode-tracking algorithm. The paper is well-structured.
•	The experimental results in simulation are strong.

Weaknesses:
•	Although the experimental results are state-of-the-art, the paper lacks evaluation on real-world applications. This is a notable gap, especially given that practical utility in real-world robotic control is more important than performance in simulation and that other baselines include such experiments.
•	The authors mention that mode tracking “introduces a minor overhead compared to the transformer forward pass,” but an inference time comparison experiment is missing. Including such an analysis would help substantiate this claim.
•	 The novelty is somewhat limited, as the GIVT model architecture is adopted from another domain, and the mean-shift algorithm is not new.

---

> ### Author Rebuttal · Authors · 2025-07-31
>
> Thanks for your thorough evaluation and insightful feedback on our work. We agree that demonstrating practical utility and providing further analysis is important. To address the concerns raised, within the short period of the rebuttal, we have conducted new experiments and will add several key clarifications to the manuscript.
>
> -  **Autonomous Driving Experiments**: We have conducted new experiments on the nuScenes autonomous driving benchmark [Caesar et. al, 2020]. Our preliminary results show that Q-FAT achieves performance competitive with the state-of-the-art VQ-BeT baseline, demonstrating that our method is effective in complex, real-world-adjacent scenarios. Here is the table with preliminary results from our initial experiments (lower is better):
>
>     | Method       | L2 Error (m) | Collision Rate (%) |
>     | :----------- | :----------- | :----------------- |
>     | Diffusion    | 0.96         | 0.44               |
>     | VQ-BeT       | **0.73**         | **0.29**               |
>     | Q-FAT (ours) | 0.75         | 0.37               |
>
>     We plan to add this experiment to our revised manuscript.
>
> -  **Inference Time Analysis**: We have performed an inference time comparison between the sampling algorithms on a trained model from the Kitchen environment, where the model has 4 mixture components. Here is the table of the results:
>
>     | Method        | Time (milliseconds) |
>     | :------------ | :------------------ |
>     | GMM Sampling  | 1.061 ± 0.125   |
>     | Mode Sampling | 1.550 ± 0.314   |
>
>     The performance was tested on a 16 GB Macbook Pro CPU. We will add a more comprehensive study in the revised manuscript across the different environments.
>
> - **Demonstration Data Collection**: We apologize for the omission. The datasets used are standard benchmarks in the field, collected via a mix of human teleoperation and scripted expert policies, consistent with the methods used for our baselines [Chi et. al 2023, Lee et. al 2024]. We will add a dedicated paragraph in the appendix clarifying the source and collection method for each environment's demonstration data. All the data are expert level trajectories, since we are in a behavioral cloning setup.
>
> ---
> ### References
> [Chi et. al 2023] Diffusion Policy: Visuomotor Policy Learning via Action Diffusion
> [Lee et. al 2024] Behavior Generation with Latent Actions
> [Tschannen et. al 2024] JetFormer: An Autoregressive Generative Model of Raw Images and Text

---

> > ### Comment · Reviewer_DqgJ · 2025-08-07
> > **Concerns partially addressed**
> >
> > My question was about the inference time comparison across baselines, however the rebuttal only talks about the comparison of the sampling.
> >
> > Meanwhile, the inference speed advantage is not significant. The inference time for one of the baselines, VQ-BeT, is almost 20 ms (15 ms for conditional) in the same environment (see attached figure). In addition, the performance on the autonomous driving system is not good enough compared to the baseline.

---

> > > ### Author Response · Authors · 2025-08-07
> > >
> > > Thanks once again for your feedback. We further provide some clarifications below:
> > >
> > >
> > > - **Inference time**: The inference time experiments across baselines has already been performed in the VQ-BeT paper (Section 4.4, Figure 3). Our model and VQ-BeT share the same backbone model (minGPT) and we expect only a slight increase in inference speed to our model (due to the absence of a separate quantization network). Thus, we wanted to address your main concern about the overhead of the mode sampler, for which we provided extra experiments. We are happy to provide a more thorough inference time discussion in our revised manuscript.
> > >
> > > - **Autonomous driving experiment**: While our results are preliminary, our model is the second best, compared to the baselines (Trajectory Diffusion, VQ-BeT and GPTDriver). We would like to further highlight the fact that the L2 error metric indicates an extra 2cm error compared to VQ-BeT on average, which is marginal. Due to time limitations, we were not able to perform hyper-parameter tuning, however our results strongly indicate that our model can perform on par with VQ-BeT in such a complex environment.
> > >
> > >
> > > We are confident that with our clarifications, your concerns should be fully addressed.

---

> > > ### Author Response · Authors · 2025-08-08
> > > **Inference time comparison**
> > >
> > > Upon reading your review again, we realized that a direct performance profile would best address your question regarding inference speed. Our analysis shows that our model is approximately **twice as fast**, with an inference time of *~1.1 ms* (GMM sampling) compared to VQ-BeT's *~2.2 ms*:
> > >
> > > * **Identical Backbone:** Both models spend a similar amount of time on the minGPT backbone, approximately **1.0 ms**.
> > >
> > > * **Action Decoding:** The primary difference lies in the action decoding heads.
> > >     * VQ-BeT utilizes two separate 2-layer MLPs to decode the action code and a corrective offset, which takes approximately **1.0 ms**. The overhead from the VQ layers is minimal (**~0.2 ms**).
> > >     * In contrast, our model uses a single linear layer to output the GMM parameters, which only takes **~0.1 ms**.
> > >
> > > The performance was tested on a 16 GB Macbook Pro CPU.

---

### Official Review · Reviewer_vHHX · 2025-07-02

**Clarity:** 3
**Significance:** 4
**Originality:** 4
**Rating:** 5
**Confidence:** 3

**Summary:**

In this work, Q-FAT is introduced as an alternative to action models that are either diffusion based (thus slow), and autoregressive-based (thus relying on vector quantization). The approach is based on a recent Generative Infinite-Vocabulary Transformer (GVIT), which allows for continuous parametrization through a mixture of gaussians. This allows for quick inference and for sampling to happen, which is essential when the outputs of the model must be used as action policy. In the work, two sampling techniques are explored to reduce jitteriness in the predicted trajectory, showing that both are successful. The experiments are on 9 different tasks, where the method display state-of-the-art performance.

**Questions:**

How would you address the scaling question? This model is transformer-based, however, it relies on GMM. Will this hinder the scaling performance?

Please alter the plots to have larger labels, and improve the clarity of Figure 1.

**Ethical Concerns:**

["NO or VERY MINOR ethics concerns only"]

**Final Justification:**

This work has shown a novel and successful application of GVIT to robotics, which will be useful to the community.
The methods are mostly well described and the main concerns were addressed in the rebuttal. The results are positive and support the claims. This is why I believe this paper should be accepted.

**Limitations:**

Yes

**Paper Formatting Concerns:**

No formatting concerns.

**Quality:**

3

**Strengths And Weaknesses:**

**Quality**. The methodology is well-introduced and justified, by explaining the downsides of previous methods. The issues with sampling strategies are exemplified, and later explored in the experiments. The model is evaluated on diverse tasks, showing good performance against relevant competing approaches.

Given the similar performance of the two sampling techniques it is not clear to me why they are both introduced. What are the true advantages of one over the other? Could this be demonstrated, or at least explained more clearly?

The issue of the scalability of such approach is also important and is not addressed.

**Clarity**. The manuscript is clear and well written. The metrics are well-explained and it is clear to see when the method outperforms the others, and also by how much. Model behavior is exemplified with plots.

The plots should have larger labels, that are consistent with each other.
The first figure could be more clear, the exact environments are not clear, and the masking used in the transformer is also not specified. Finally, for Figure 1, it would be important to also highlight that special sampling techniques are used to get smooth trajectories.

**Significance**. This is a significant contribution to the field. The model proposed addresses specific issues, and could be used as inspiration for further development, for example when combined with LLMs.

**Originality**. The proposed approach takes a previously-proposed model, and adapts it in an original way for robotic applications. This is an original piece of work.

---

> ### Author Rebuttal · Authors · 2025-07-31
>
> Thanks for your thorough evaluation and insightful feedback on our work. We will update the description of Figure 1 as you suggested. We further answer your questions below:
>
> - **Motivation for Two Sampling Techniques**: The two techniques were introduced to address trajectory 'jitter' that we observed in preliminary experiments, even after significantly downscaling the GMM variance. This jitter arose from the model rapidly switching between different predictive modes between timesteps. Mean-shift sampling enforces temporal consistency and produces smoother trajectories, as shown qualitatively in Figures 4 and 5. We will clarify this motivation in the revised text.
>
> - **Scalability**: In terms of model and data scalability, autoregressive transformers are the most scalable model known to date, as demonstrated by frontier level LLMs. As for the GMM, the GMM head's parameter count scales linearly with the action dimension when using a standard diagonal covariance matrix, meaning it does not hinder the model's application to higher-dimensional problems. Furthermore, recent work [Tschannen et. al 2024] has demonstrated scaling GMMs to high dimensional image modeling with an autoregressive transformer (in a 128-dimensional latent space with 1k mixture components), demonstrating high sample fidelity. This indicates that scaling GMMs to complex distributions in high-dimensional spaces, such as action spaces, is straightforward.
>
> - **Masking**: We use causal masking as described in appendix D. We will highlight this more in our revised manuscript.
>
> ---
> ### References
> [Tschannen et. al 2024] JetFormer: An Autoregressive Generative Model of Raw Images and Text

---

> > ### Comment · Reviewer_vHHX · 2025-08-01
> >
> > Thank you for addressing my concerns. Looking forward to seeing the follow-up works!

---

### Official Review · Reviewer_j3Wb · 2025-07-14

**Clarity:** 3
**Significance:** 3
**Originality:** 3
**Rating:** 5
**Confidence:** 4

**Summary:**

This paper introduces Quantization-Free Autoregressive Action Transformer (Q-FAT), a novel policy representation for continuous-control imitation learning that avoids the common step of discretizing actions. Instead of tokenizing actions via VQ-VAE or k-means clustering, Q-FAT directly predicts the parameters of a Gaussian mixture model (GMM) at each timestep with a decoder-only transformer backbone. The policy outputs mixture weights, means, and (diagonal) variances, enabling exact likelihood evaluation, autoregressive sampling, and principled variance-reduction techniques (variance scaling and mode-tracking via mean-shift). Empirically, Q-FAT matches or exceeds the state of the art on nine simulated robotics tasks (e.g., PushT, Kitchen, UR3 BlockPush, Multimodal Ant), improving both success rates and behavioral diversity while maintaining efficient inference. The authors also analyze robustness to the number of mixtures, illustrate pruning dynamics, and discuss practical training details (history masking, ResNet encoder for vision, initialization).

**Questions:**

1. Could you isolate the benefit of continuous GMM outputs by comparing a discrete-token transformer with identical depth/width to Q-FAT?

2. How does Q-FAT perform on real-robot or higher-dimensional (e.g., 17-DoF humanoid) settings? Can the GMM scale gracefully as action dimensionality grows?

3. What is the wall-clock overhead of mean-shift sampling in practice? A small timing study (e.g., ms per step) would help assess real-time applicability.

4. How sensitive are the sampling techniques to varying noise levels in demonstrations? An ablation where dataset noise is artificially increased could inform robustness.

**Ethical Concerns:**

["NO or VERY MINOR ethics concerns only"]

**Limitations:**

Yes.

**Paper Formatting Concerns:**

None.

**Quality:**

4

**Strengths And Weaknesses:**

# Quality

**Strength**: solid experimental validation across diverse benchmarks, with thorough comparisons to transformer- and diffusion-based baselines (BeT, VQ-BeT, DiffusionPolicy, BESO); state-of-the-art task success rates and entropy metrics demonstrate both accuracy and multimodality.

**Weakness**: limited ablation on the impact of transformer depth/width; hard to disentangle gains from architecture vs. continuous parametrization; no real-robot or higher-dimensional continuous tasks (e.g., humanoid control) to test scalability beyond simulation.

# Clarity

**Strength**: well-structured presentation: clear problem motivation, method description, and algorithmic details (Equations 1-5, Algorithm 1); helpful figures (GMM sampling artifacts, mode pruning evolution).

**Weakness**: appendices are dense: readers may struggle to find key hyperparameters or environment details without deeply navigating Appendix D; limited discussion connecting latent continuous policies to recent diffusion-based approaches beyond performance tables.

# Significance

**Strength**: eliminates a major pipeline complexity (action quantization), simplifying training and deployment; enables exact likelihoods and fast, single-pass inference, addressing diffusion-policy latency issues.

**Weakness**: empirical gains, while consistent, are incremental (e.g., +2-7 % success), raising questions about broad impact across more challenging domains; the practical value of variance-reduction techniques might be context-specific (noisy vs. clean data).

# Originality

**Strength**: novel adaptation of Generative Infinite-Vocabulary Transformers (GIVT) to continuous control, directly modeling GMM outputs; introduction of mean-shift mode sampling in action transformers.

---

> ### Author Rebuttal · Authors · 2025-07-31
>
> Thanks for your thorough evaluation and insightful feedback on our work. We will update the appendix in the updated manuscript to make it less dense as you suggested. We further answer your questions below:
>
> - **Isolating the benefit of continuous GMM outputs by comparing a discrete-token transformer with identical depth/width**: We have indeed isolated the benefit of the continuous GMM parameterization by ensuring that we have the same or less trainable parameters than the discrete-token transformer baseline (VQ-BeT). Having an identical width and depth would give our model a disadvantage, since it does not have a separate quantization network as in VQ-BeT.
>
> - **Performance beyond simulated tasks**: We have conducted new experiments on the nuScene [Caesar et. al, 2020] autonomous driving benchmark as suggested by reviewer DqgJ. Our preliminary results show that Q-FAT achieves performance competitive with the state-of-the-art VQ-BeT baseline, demonstrating that our method is effective in complex, real-world-adjacent scenarios. Here is the table with preliminary results from our initial experiments (lower is better):
> | **Method** | **L2 Error (m)** | **Collision Rate (%)** |
> | :----------- | :--------------- | :--------------------- |
> | Diffusion | 0.96 | 0.44 |
> | VQ-BeT | **0.73** | **0.29** |
> | Q-FAT (ours) | 0.75 | 0.37 |
>
>  We plan to add this experiment to our revised manuscript.
>
> - **GMM scaling to higher dimensions**: The GMM head's parameter count scales linearly with the action dimension when using a standard diagonal covariance matrix, meaning it does not hinder the model's application to higher-dimensional problems. Furthermore, recent work [Tschannen et. al 2024] has demonstrated scaling GMMs to high dimensional image modeling with an autoregressive transformer (in a 128-dimensional latent space with 1k mixture components), demonstrating high sample fidelity. This indicates that scaling GMMs to complex distributions in high-dimensional spaces, such as action spaces, is straightforward.
>
> - **Inference Time Analysis**: We have performed an inference time comparison between the sampling algorithms on a trained model from the Kitchen environment, where the model has 4 mixture components. Here is the table of the results:
> | **Method** | **Time (milliseconds)** |
> | :------------ | :---------------------- |
> | GMM Sampling | 1.061 ± 0.125 |
> | Mode Sampling | 1.550 ± 0.314 |
>
>  The performance was tested on a 16 GB Macbook Pro CPU. We will add a more comprehensive study in the revised manuscript across the different environments.
>
> - **Sensitivity of the sampling techniques to varying noise levels in demonstrations**: Figures 4 and 5 illustrate the impact of various sampling techniques. In low-noise environments, all techniques exhibit comparable performance. However, with standard GMM sampling, an increase in noise tends to significantly destabilize the generated paths. For variance-reduction and modes sampling, the quality of actions is highly contingent on the underlying structure of the ground truth distribution and the number of mixtures used to model it, which we visually represented in Figure 2.
>
> --------------------------------------
> - [Caesar et. al, 2020] nuScenes: A Multimodal Dataset for Autonomous Driving
> - [Tschannen et. al 2024] JetFormer: An Autoregressive Generative Model of Raw Images and Text

---

> > ### Comment · Reviewer_j3Wb · 2025-08-07
> >
> > Thank you for the detailed rebuttal which has addressed all my concerns.

---

### Decision · Program_Chairs · 2025-09-17

**Decision:**

Accept (spotlight)

**Comment:**

The paper introduces Q-FAT, a quantization-free autoregressive transformer that directly parameterizes a GMM over actions, simplifying the imitation-learning pipeline while achieving strong performance across diverse simulated robotics tasks. Reviewers highlight clear motivation, thorough experiments (nine tasks), and practical sampling strategies (variance scaling and mean-shift) that reduce trajectory jitter, with two reviews recommending accept and a third moving to borderline accept after rebuttal. The rebuttal also provides preliminary nuScenes results and a concrete inference-time breakdown supporting efficiency claims. The review ratings are all positive. Thus, the AC decides to accept the submission.